# Depression, Olfaction, and Quality of Life: A Mutual Relationship

**DOI:** 10.3390/brainsci8050080

**Published:** 2018-05-04

**Authors:** Marion Rochet, Wissam El-Hage, Sami Richa, François Kazour, Boriana Atanasova

**Affiliations:** 1UMR 1253, iBrain, Université de Tours, Inserm, 37200 Tours, France; marion.rochet@etu.univ-tours.fr (M.R.); wissam.elhage@univ-tours.fr (W.E.-H.); francoiskazour@hotmail.com (F.K.); 2CHRU de Tours, Clinique Psychiatrique Universitaire, 37044 Tours, France; 3Department of Psychiatry, Faculty of Medicine, Saint-Joseph University, P.O. Box 17-5208, 11-5076 Beirut, Lebanon; richasami@hotmail.com; 4Psychiatric Hospital of the Cross, 60096 Jal Eddib, Lebanon

**Keywords:** depression, olfaction, markers, quality of life, therapeutic tool

## Abstract

Olfactory dysfunction has been well studied in depression. Common brain areas are involved in depression and in the olfactory process, suggesting that olfactory impairments may constitute potential markers of this disorder. Olfactory markers of depression can be either state (present only in symptomatic phases) or trait (persistent after symptomatic remission) markers. This study presents the etiology of depression, the anatomical links between olfaction and depression, and a literature review of different olfactory markers of depression. Several studies have also shown that olfactory impairment affects the quality of life and that olfactory disorders can affect daily life and may be lead to depression. Thus, this study discusses the links between olfactory processing, depression, and quality of life. Finally, olfaction is an innovative research field that may constitute a new therapeutic tool for the treatment of depression.

## 1. Introduction

Depression and olfactory dysfunction induce long-term impairments that affect individuals and have a major impact on subjects’ social skills, relationships, wellbeing, and quality of life [1,2]. Major depression is one of the two most debilitating diseases according to the World Health Organization (2010). Depression affects 8 to 12% of the world’s population [3]. In France, 9 million people experience at least one depressive episode during their lifetime (INPES, National Institute for Prevention and Health Education). Olfactory dysfunctions are present in 22% of individuals aged between 25 and 75 years old [4]. In these situations, the impact of this olfactory impairment on patients’ life is often neglected. However, olfaction provides people with valuable input from the chemical environment around them. Smell can also impact different psychological aspects of the subject’s life by forming positive and negative emotional memories related to smell. It can also affect the social abilities and the interpersonal relationships of the individual. When the olfactory input is distorted, disability and decreased quality of life are reported [5].

In the last two decades, several studies have investigated olfactory dysfunction in depression. Several reasons are behind the study of olfaction in depression. First, the rapid expansion of structural and functional imagery techniques that uncover overlapping brain areas involved in olfactory processing and depression. These regions are mainly the orbitofrontal cortex, the anterior and posterior cingulate cortices, the insula, the amygdala, the hippocampus, and the thalamus [6,7,8]. Second, olfactory stimuli are encoded and may activate emotional memory. This could be explained by the close anatomical links between the olfactory system and the brain circuits involved in memory [9] and emotion [10]. These two cognitive functions are frequently altered in depression [11]. Third, olfactory dysfunction may induce symptoms of depression, due to the impact of odors on emotions, mood, or behaviors [12]. Indeed, a recent systematic review has shown a reciprocity and a close association between olfactory impairment and depression [13]. Thus, depressed patients have lower olfactory performance than healthy controls, and patients with olfactory dysfunction have symptoms of depression that worsen with the severity of olfactory impairment. Fourth, bilateral olfactory bulbectomy in rodents induces changes in behavior, as well as in the endocrine, immune, and neurotransmitter systems. These changes are similar to those seen in patients with major depression [14,15]. These alterations are reversed by antidepressants, suggesting that the dysfunctions seen in depression could be related to alterations in the olfaction system. However, these findings cannot be replicated in humans. Nevertheless, studies showed that olfactory bulb volume differs significantly between depressed and non-depressed individuals and seems to be a promising marker for depression [16]. Fifth, stress, a triggering factor of depression in vulnerable subjects, induces behavior similar to some symptoms of depression, as well as decreased cell proliferation or neurogenesis, both in the hippocampus and in the olfactive bulbs [17]. Finally, it has been suggested in human [18,19,20,21] and animal studies [22,23,24] that olfaction can be used as a therapeutic tool for depression.

The exact relationship between depression and olfactory dysfunction is still unclear, even if both disorders coexist and affect quality of life, social and professional integration, as well as family balance [25]. Several reviews have studied olfactory perception in depression [26,27], the relationship between olfactory loss and quality of life [28,29], and between depression and quality of life [1]. In the present review, we overview the links between depression, olfaction, and quality of life of patients in order to understand the relationship between olfactory function and depression, and to determine if olfactory loss affects patients’ quality of life. In addition, we discuss the possible use of olfactory stimulation as a promising therapeutic tool to potentiate the antidepressant effect of medication.

## 2. Depression

Depression is a multifactorial disease involving multiple etiologies, with the contribution of biological, genetic, and environmental factors [30].

From a biological perspective, depression is associated to a monoaminergic deficiency in the brain [31]. Indeed, the role of monoaminergic neurotransmitters (noradrenaline, serotonin, and dopamine) has been demonstrated in the control of mood and cognitive functions. Neurotransmitter-based drug treatments accelerate clinical improvement, but resistance to antidepressants implies the involvement of other etiologies. One hypothesis is the inhibition of hippocampal neurogenesis, since long-term use of antidepressants leads to an increase of adult hippocampal neurogenesis [32]. A decrease in hippocampal volume has been demonstrated in depression [33] at the earliest stages of the disease and was correlated with patients’ cognitive impairment [34]. Brain abnormalities have also been observed in the orbitofrontal cortex, anterior cingulate cortex [35], and the amygdala [36]. It has been shown that amygdala volume is reduced in untreated depressed patients, but increases with treatment [37]. 

As for the genetic etiologies of depression, studies show contradicting results. A meta-analysis suggested that depression is a familial disorder and with an estimated heritability of 31 to 42% [38]. For instance, a functional polymorphism of the serotonin transporter gene would change the impact of stressful life events on depression [39]. However, another meta-analysis was not able to show any association between the serotonin transporter genotype and depression [40]; and a genome-wide association study could not clearly identify the involvement of several genes in depression [41].

Concerning the psychological perspective, Beck [42] proposed a model based on the fact that negative life events may trigger depressive episodes. Therefore, certain life experiences can modify the psychological functioning of the individual and induce emotional instability. Kendler et al. [43] proposed the kindling-sensitization hypothesis where negative life events that were initially unable to trigger a depressive episode later gained a capacity to trigger a recurring episode.

Pathophysiology of depression is complex and remains partially elucidated. According to the latest edition of the Diagnostic and Statistical Manual (DSM-5, [44]), a major depressive episode is diagnosed, when (i) five (or more) of the following symptoms are present during the same two-week period: depressed mood most of the day, markedly diminished interest or pleasure in all, or almost all, activities most of the day, nearly every day, significant weight loss when not dieting or weight gain, insomnia or hypersomnia, psychomotor agitation or retardation, fatigue or loss of energy, feelings of worthlessness or excessive or inappropriate guilt, diminished ability to think or concentrate and recurrent thoughts of death; and (ii) a change from previous functioning with at least one of the two major symptoms of depression: depressed mood or loss of interest or pleasure. These symptoms are associated with significant clinical distress or impairment in social, occupational, or other important areas of functioning. It was shown that depression alters aspects of information processing, including perception and attention, memory processes (e.g., preferential recall of negative information rather than positive one), and interpretation of ambiguous information [11]. Recent literature explored the presence of depression-associated sensorial biases, particularly at the olfactory level by investigating the state (disappearance of olfactory alterations in clinically improved patients) and the trait (persistent olfactory alterations after clinical improvement) olfactory markers of major depression [27,45,46,47,48].

## 3. Olfaction

### 3.1. Functioning of Olfactory Perception

The olfactory information (i.e., the chemical molecule), will first settle at the level of the olfactory receptors, within the olfactory epithelium. There are two ways for volatile chemical molecules to reach the olfactory epithelium. The first is the orthonasal pathway (so-called “direct” pathway) and takes place during inspiration through the nose. The second is the retronasal pathway (so-called “indirect” pathway) and takes place during mastication through the mouth. The binding of the molecule on specific olfactory neuroreceptors, triggers an action potential on these cells [49]. The information then reaches the olfactory bulb, located at the base of the frontal lobe [2]. The olfactory bulb sends projections via the lateral olfactory tract to different brain areas, such as the olfactory tubercle, the anterior olfactory nucleus, the piriform cortex, the lateral entorhinal cortex, or the ventral tenia tectae [50], allowing olfactory information to reach different brain levels (e.g., the thalamus, hypothalamus, or hippocampus). Indeed, the olfactory system is connected to the limbic system through the amygdala, the piriform cortex, the anterior cingulate cortex, the insula, and the orbitofrontal cortex [7]. These connections may explain how odors an induce mood changes [51] and impact cognitions and behaviors [52].

### 3.2. Evaluation of Olfactory Functions

Various olfactory tests are available for the evaluation of olfactory function. Psychophysical measures have been used to establish a link between measurable parameters (i.e., product concentration, chemical composition) and the qualitative (i.e., odor’s identification, odor’s naming) and quantitative (i.e., perception of odor’s intensity) characteristics of the evoked stimulus. It is, thus, possible to evaluate parameters, such as the odor’s intensity, familiarity, or hedonic level by using scales of measurements. The psychophysical tests provide a quantitative measure of sensory function by using the verbal response of the subject as an indicator of the olfactory perception. Generally, the psychophysical tests include investigation of odor detection threshold, odor discrimination, and identification. Standardized tests have been developed to evaluate these three olfactory functions. They were validated with several hundred healthy participants (men and women) of different ages. Doty et al. [53] has developed the University of Pennsylvania Smell Identification Test (UPSIT), an olfactory evaluation tool in the form of a scratch ’n sniff test. This test can measure individuals’ odor identification ability. The Sniffin’ Stick Test created by Hummel et al. [54] evaluates odor detection threshold, odor discrimination, and identification ability. This test is formed of several odorous pens (sticks). For the olfactory threshold test, stick triplets with increasing odor concentrations are presented to the subject. In every triplet, only one stick contains the odorant (2-phenyl ethyl alcohol). The subject is asked to identify in every triplet the pen that contains the odorant, the other two being without odor. The discrimination test is done with the following presentation of 16 odorous pen triplets. In this test, the subject has to detect if the pen with the odorant is different from the other two. Finally, during the odor identification test, 12 or 16 odorous pens are presented to the subject. Using a multiple forced-choice task, odors are identified from a list of four descriptors for each odor. A final score evaluates the overall olfactory capacity of the subject and determines whether the individual is normosmic, hyposmic, or anosmic.

Occasionally, devices called olfactometers are used to carry out the psychophysical measures while providing a better control of the delivery conditions of the odorant. It allows the delivery of a precise concentration of the odorant at the entrance of the nostrils via a controlled airflow. It is also possible with this device to have either unilateral or bilateral nostril stimulations. This tool seems to be more appropriate in situations where the control of odors’ concentration is important, like in olfactory threshold measures.

The psychophysical tests described above are easy to administer, to transport, are not expensive, and provide a rapid time of analysis (between 5 and 40 min depending on the test). They are the most used in the clinical field. However, the psychophysical evaluation of the olfactory functions demands an active participation of the subject. To avoid this inconvenience neurophysiological techniques are used to measure the human electrophysiologic response to an odorant stimulus. They include odor event-related potentials (OERPs) and a summated potential recorded from the surface of the olfactory epithelium (the electro-olfactogram, EOG). Many other techniques have also been developed, including electroencephalographic testing, measurement of psychogalvanic skin response to olfactory stimuli, measurement of respiratory, cardiovascular, papillary, and oculomotor reflexes [55]. More recently, structural and functional imaging technologies (positron emission tomography, single photon emission tomography, and high-resolution structural magnetic resonance imaging) were developed in order to study the central pattern associated to the olfactory perception.

In summary, a number of techniques and tools are available to explore the olfactory function, each having its own advantages and disadvantages. It is important to note that all these methods (psychophysical, neurophysiological, and neuroimaging techniques) are complementary because they do not measure the same olfactory parameters, function, and process. Some authors described and compared the existing methods and techniques (see [55,56]).

The methods described above can be used to study olfactory function and to detect the presence of olfactory disorders. Olfactory disorders are divided into quantitative and qualitative disorders. Quantitative olfactory disorders are hyposmia, hyperosmia, or anosmia [50]. They are defined, respectively, as a decrease, an increase, or a complete loss of olfactory perception. Qualitative olfactory disorders are parosmia (bad perception of smell) and fantosmia (an olfactory hallucination, or perception of odors that are not present within the olfactory field) [57]. Two thirds of cases of anosmia and hyposmia are due to either an upper respiratory tract infection, a brain trauma, or to nasal pathologies that damage the olfactory neuroepithelium [58]. Moreover, it has been demonstrated that olfactory perception is altered in patients with neurodegenerative diseases. Indeed, olfactory disorders are seen in 85 to 90% of patients with Alzheimer and Parkinson diseases. However, these patients are rarely aware of their olfactory impairment.

## 4. What Is the Link between Olfaction and Depression?

### 4.1. Anatomical Link

Several brain areas play a role in olfactory perception and are involved in the etiology of depression. First, the olfactory bulb transmits olfactory information to other brain areas, like the amygdala, the hippocampus, and the anterior angular cortex [50]. It has been shown that bilateral olfactory bulbectomy in rodents causes changes to the immune and endocrine systems similar to those seen in depression [15]. Indeed, bilateral destruction of olfactory bulbs leads to alteration in serotonin and dopamine concentration [14]. Bilateral olfactory bulbectomy can also induce behavioral changes, such as a reduction in sexual behavior [59], an odd food-motivated behavior [60], a decrease in anxiety-like behavior [61], and an increase in depression-like behavior [15,62]. In addition, a study found a reduced volume of the olfactory bulb in depressed patients [63]. A study using a rat model of depression, with unpredictable chronic mild stress (UCMS), has observed a reduced amount of olfactory receptor neurons in olfactory epithelium [64]. These results may explain some of the impairment in olfactory sensitivity observed in depressed patients.

A recent study showed that in a murine model that mimics the hyperactivation of the HPA axis (and thus causes a phenotype of depression), olfactory function, and also adult neurogenesis at the subventricular zone and the dentate gyrus, were affected [48]. In this study, mice receiving chronic administration of corticosterone had deficits in their olfactory acuity, fine odor discrimination, and olfactory memory. In addition, cell proliferation in the subventricular zone and the dentate gyrus (two niches of adult neurogenesis), and the survival of new neurons in the dentate gyrus and olfactory bulbs, were decreased by corticosterone administration. Antidepressant treatment (fluoxetine) allowed a return to normal olfactory function and adult neurogenesis [48]. Therefore, the link between olfaction and depression can also be explained by the decrease in neurogenesis.

Other areas, such as the amygdala or hippocampus, also have a role in olfaction and depression. Indeed, the hippocampus is involved in odor storage tasks [65] and in depressive symptoms, such as deficits in autobiographical memory [66]. In addition, studies have shown decreases in hippocampal volume associated to depression [33]. It has been shown that the amygdala of healthy individuals is activated during the evaluation of intensity, hedonic aspect and memory of odor-related emotions [67]. The amygdala would be hyper-activated in depression [68].

The orbitofrontal cortex is also implicated in the link between olfaction and depression. It is involved in attention, emotional, and cognitive processes of depression. On one hand, the ventromedial part is involved in rumination, anxiety, and sensitivity to pain, and is hyper-activated in depressed patients. On the other hand, the dorsal part is involved in psychomotor retardation, apathy, attention disorders, and working memory, and hypoactive in depressed patients [69]. The role of this cortex is crucial in olfaction, but is still controversial in depression [70,71]. Some authors consider that an unpleasant stimulus activates the left part of this cortex and a pleasant stimulus activates the right side [71], but other authors have shown that the activation of the orbitofrontal cortex is not a function of positive or negative valence of odorants [72]. The orbitofrontal cortex is involved in the identification of odors and in olfactory memorization [71,73].

The cingulate cortex is involved in both olfactory function and depression. In depression, the volume of its anterior part is diminished [35,74]. This would be partly responsible for the increased recurrence of depressive episodes [75]. As for its role in olfaction, the activation of this brain area depends on the hedonic valence of the odor [76]. 

The insula participates in the evaluation of emotional states and more particularly of the bodily sensations during an emotional experience [77]. A study showed that the insular cortex contributes to odor quality coding by representing the taste-like aspects of food odors [78]. The insula has higher levels of activity in resting states, increasing the inability of depressed patients to disengage from externally-cued events, and leading to pathological self-focused mental ruminative behaviors [79].

Finally, the habenula is affected by olfactory bulb input and is involved in the regulation of psychomotor and psychosocial behaviors [57]. Its metabolic activity is increased in animal models of depression [80]. The role of the habenula is the transfer of olfactory information to other brain areas [81]. It is activated in response to emotionally-negative stimuli [82]. According to Oral et al. [83] habenula plays a decisive role in the link between olfactory disorders and depression since a bilateral bulbectomy induces a structural degeneration by apoptosis of the habenula, leading to the appearance of the main symptoms of depression. A summary of the brains areas involved in olfaction and depression is presented in Table 1.

### 4.2. Olfaction: A Marker of Depression?

Over the years, several studies have focused on the impact of depression on olfactory functions like detection thresholds, identification, and discrimination capabilities, and the assessment of hedonicity or intensity. The strong link between depression and olfaction has allowed researchers to espouse the hypothesis that a reduced olfactory capacity may be a marker of depression [45,84]. Two types of olfactory markers have been proposed: (i) the state olfactory markers where olfactory impairments disappear after antidepressant treatment; and (ii) the trait olfactory marker, where olfactory impairments persist after clinical remission. 

Most studies have shown that the detection threshold of depressed subjects was increased compared to controls [27,63,85,86]. However, some authors reported an unchanged olfactory threshold [87,88,89]. Few studies investigated the olfactory threshold in remitted patients after antidepressant treatment and showed conflicting results. Gross-Isseroff et al. [90] have demonstrated an increase of olfactory odor sensitivity in remitted patients suggesting that this could be due to antidepressant treatment. Another study observed a significant negative correlation between olfactory sensitivity and depressive symptoms [91]. Pause et al. [86] reported remission of odor threshold impairment in depressed patients after antidepressant treatment. All these observations suggest that the reduced olfactory sensitivity may be a marker of depression. However, further studies are needed to confirm whether this olfactory function is restored by antidepressant treatment.

Some studies have shown that depression is associated with a lower olfactory identification capacity [92,93,94]. Two studies reported that depressed subjects had a lower identification capacity of the components of a complex odorant environment (binary iso-intense mixtures), during the major depressive episode [46,95]. Naudin et al. [46] also showed that the olfactory alteration persists after clinical improvement reflecting olfactory trait markers of depression. The majority of studies have demonstrated that olfactory identification capacities are not altered in depression [46,85,87,88,89,91,96,97,98,99,100]. In summary, odor identification function didn’t seem to be altered in depression when standardized olfactory tests were used. Moreover, it has been proposed that an odor identification parameter could be used to differentiate between depressed patients and Alzheimer’s disease (AD) patients since comparative studies showed that this function is altered in AD, but not in depression (for review, see [8]).

Concerning odor hedonic perception, only two studies have shown unchanged scores between depressed patients and healthy controls [89] before or after antidepressant treatment [92]. For the majority of the investigations, this parameter is influenced by depressive state [46,47,85,86,95,101]. Atanasova et al. [95] showed that depressed patients would perceive unpleasant odors as more unpleasant (negative olfactory alliesthesia), while pleasant odorants would be perceived as less pleasant (olfactory anhedonia) in comparison to controls. Naudin et al. [46] reported that this hedonic olfactory bias concerns a highly emotional odorant and that it vanishes after antidepressant treatment. Therefore, it is considered as an olfactory state marker of depression. In contrast to these observations, other studies revealed that depressed patients over-evaluate the hedonic perception of odors [85,86]. Lombion-Pouthier et al. [85] suggested that this over-evaluation could be due to modifications in the orbitofrontal cortex observed in depression (this structure is also applied in hedonic perception of odor). 

Several studies investigated changes in olfactory intensity rating in depression [85,86,92,95,102,103]. Only one showed that depressed patients perceived pleasant stimuli as less intense and unpleasant stimuli as more intense than controls [95]. However, the majority of the studies did not find any associations between that hedonic value of odors and changes in the perception of odor intensity.

As for the perception of familiarity, studies show contradictory results. Some authors could not find any change in familiarity ratings associated with depression [47,102,103], whereas others found lower familiarity ratings in depressed patients compared to controls [46]. Future studies are needed to clarify this aspect of olfaction. The edibility perception of odors has not been studied in depression. However, knowing that eating disorders are frequently observed in depression, this parameter should be a subject for future investigation.

Studies show that odor discrimination abilities are not different in depressed patients compared to controls [63,84,92,95]. These same studies found similar results with respect to odor intensity assessment capabilities. A summary of the studies exploring the deficits in the different olfactory functions in depressed patients and in clinically-improved patients are presented in Table 2.

Olfactory impairments were also described in other psychiatric, disorders like schizophrenia [107], bipolar disorder [108], or posttraumatic stress disorder [109]. However, there are very few studies comparing olfactory dysfunction in depression and other affective disorders. When assessing olfactory thresholds and identification abilities, Swiecicki et al. [89] could not find any difference between patients with unipolar and bipolar depression. However, a recent study [106], showed that impairment in odor identification may be seen in mood disorders (major depressive and bipolar I disorder), with a more pronounced impairment associated to psychotic depression features. On the other hand, global olfactory dysfunction observed in schizophrenia may not be a feature of other neuropsychiatric conditions [106]. Olfactory markers of differentiation were also described in depression and Alzheimer’s disease [8]. Future investigations are needed in order to investigate potential markers of differentiation between other neuropsychiatric disorders.

## 5. Quality of Life: Definition and Evaluation

The World Health Organization, defined the quality of life as “the perception of an individual of his place in existence, in the context of the culture and the system of values in which he lives, in relation to its objectives, expectations, norms and concerns. It is a very broad concept influenced in a complex way by the physical health of the subject, his psychological state, his level of independence, his social relations as well as his relation to the essential elements of its environment”. Questionnaires have been created to assess the quality of life, like the Questionnaire of Olfactory Disorders [25]. This questionnaire was developed to assess the impact of olfactory disorders in daily life and is composed of 52 statements divided into three areas: (1) “negative statement”, which provides information on the degree of suffering of the respondent; (2) “positive statement”, that gives information on the way respondents cope with their olfactory disorders; and (3) “socially desired”. This third area allows the evaluator to assess the credibility of participants’ responses. Indeed, it helps knowing if the participant is trying to give a certain impression or socially desirable answers.

Other questionnaires used to assess quality of life include the “Short Form 36 Health Survey”, “The General Well Being Schedule” [110], “The 90-item symptom checklist” [111], and “The Nottingham Health Profile” [112]. In these different questionnaires, several areas, like general health, vitality, depression, anxiety, sleep, and pain are evaluated. However, sensory functions, like taste and olfaction, are not evaluated. A study by MacDowell et al. [113] compared the reliability, validity, and duration needed for several of these instruments. They concluded that most of these tools perform adequately for survey research purposes. 

This is a non-exhaustive list of standardized questionnaires used to assess quality of life, but researchers may prefer to use other non-standardized questionnaires targeting specific fields of research (olfactory disorders, etc.). For example, some studies used a self-report where participants had to write their feeling about their loss of olfaction [5,114].

## 6. Impact of Olfactory Disorders on Quality of Life

It is interesting to focus on the impact of olfactory loss (partial or total, reversible or not) on the diet in general. Individuals with olfactory disorders report having a more complicated alimentation because of the important link between smell and taste. Patients with olfactory disorders (69%, n = 239) had lower ratings of food since the beginning of their disorder [115]. This lower rating leads to a decrease in appetite in 56% (n = 278) [114], 32% (n = 72) [116] and 27% of subjects (n = 50) [117]. All the subjects included in these studies had olfactory disorders and the results were obtained through self-reporting questionnaires and the Multi-Clinic Smell and Taste Questionnaire [118]. 

Individuals sometimes develop coping strategies: they can increase the taste by adding salt, sugar, or spices [5,28]. Moreover, 3 to 20% of individuals with olfactory disorders report eating more than before, while from 20 to 36% report eating less [115]. The presence of olfactory dysfunction causes difficulty in maintaining a healthy balanced diet. Individuals with olfactory disorders also have difficulties cooking food because they have difficulties to detect burning food. They may burn or stale food because of their olfactory impairment [5,114,119,120]. They also have difficulties to detect gas leaks or smoke odors [120].

In addition, patients are no longer able to detect their own body odor (sweating, bad breath) [114] and report problems of hygiene. This can lead in some cases to excessive showers in order to have a higher self-confidence. For some individuals, this is the most negative effect of olfactory loss [116,117]. Between 1/4 and 1/3 of patients with olfactory disorders report having social difficulties related to hygiene problems [117,121]. Indeed, an individual with olfactory disorders is socially vulnerable if he is not able to smell his body odor or odors of others. This places him in an uncomfortable situation and increases the risk of social isolation.

Olfactory disturbance interferes with professional life in 3 to 8% of cases [116,117]. Temmel et al. [114] reported that 8% of participants with olfactory disorders had problems in their professional lives. The impact of an olfactory disorder on professional life depends on the type profession: people who work in oenology, in gastronomy, in the perfume industry, or even nurses or firefighters, can have major impairments in their professional lives [28]. A loss of smell can also be problematic in this kind of situation and add major concerns for the future professional. All of this can lead to anxiety, mood disturbances, and depression [58]. Olfactory disorders that cause food disturbance (whether cooking or eating) can also constitute factors that affect mood and lead to anhedonia [28]. 

Several studies have shown the impact of olfaction on social and family relations [122,123] such as the mother-child bond [124] and men-women relationships (i.e., in reproduction, avoidance of consanguinity, selection of partner) [125,126]. For example, olfaction is involved in the detection of fear signals. In a study by Ackerl et al. [122], women watched a terrifying movie while wearing axillary pads. A neutral film was presented the next day as control. Other women then had to smell and categorize the axillary pads obtained after the presentation of the terrifying film and those obtained after the presentation of the neutral film. As a result, women were able to distinguish between the pads of fear and those of non-fear. These results assume the existence of an odor of fear, present in perspiration and detectable by other individuals. Such behaviors may be also affected by olfactory disorders, but further studies still need to be done in this area.

## 7. Olfaction: A New Therapeutic Tool?

### 7.1. Animal Model and Odorants

Studies in animals have shown effects of odors on the emotional state. Komiya et al. [24] have demonstrated the anti-stress effects of lemon oil vapor on mice during behavioral tests (elevated plus maze and forced swim tests). The mechanism hypothesis is that lemon essential oil vapor affects the response to dopaminergic activity by modulating serotoninergic activity and/or GABA-benzodiazepines receptor complex.

In another study, Xu et al. [23] showed the effect of vanillin on the depressive-like behavior of non-bulbectomized and bulbectomized groups of rats. This was a comparative study of the effects of vanillin on depressive-like behavior rats induces by two ways: chronic unpredictable mild stress (CUMS) and olfactory bulbectomy. Results showed a significant decrease of depressive-like behavior for the CUMS group exposed to vanillin and the CUMS group exposed to fluoxetine. No changes were observed for the bulbectomy group exposed to vanillin. These results suggest that vanillin may have an effect on the symptoms of depression if the olfactory pathway is intact [23]. A study on Mongolian gerbil (which has more neuroendocrine similarity with humans than mice and rats) highlighted the anxiolytic effect of lavender odor. The authors showed that after chronic exposure to this odor, the anxiolytic effects were almost similar to those obtained after exposure to diazepam [22], but the mechanisms were not completely elucidated. Another study on rats demonstrated the antidepressant effect of lemon odor during behavioral tests (forced swim test and open field test) [127]. Lemon odor significantly reduced the depressive-like behavior. Further studies on the effects of odors on affective states in animal models are needed in order to understand the underlying mechanisms of these effects.

However, it is important to note that the cited studies have used bulbectomy to develop an animal model of depression. Their validity was, however criticized (for review, see [128]). Therefore, it is important to be careful before applying the findings obtained in animal models onto human subjects. 

### 7.2. Use of Odors in Humans

Some studies have highlighted the sedative [129] and the anxiolytic effects [130,131] of odors in humans. Lehrner et al. [129] showed that participants exposed to the ambient odors of orange essential oils had lower levels of anxiety compared to control participants. The same authors extended their observations by using lavender odor [131]. The objective of this last study was to compare the effect of odor to the effect of music in the waiting room of a dental office. The results showed that the lavender odor had a more pronounced anxiolytic effect compared to music and control conditions. Indeed, studies showed that lavender acts post-synaptically and modulates the activity of cyclic adenosine monophosphate (cAMP). A reduction in cyclic adenosine monophosphate activity is associated with sedation [132]. 

Studies in humans have highlighted the interest of using odors as therapeutic tool. In a study by Hummel et al. [21], participants with olfactory loss (various origins: post-infectious, post-traumatic or idiopathic) followed a 12-week olfactory training program. At the end of this training, measurements of olfactory function were performed and it appeared that repeated daily olfactory stimulation improved the olfactory function of participants.

Haehner et al. [19] has proposed olfactory training in patients with Parkinson’s disease. They used the same protocol as Hummel et al. [21]. The results of the study showed that olfactory training produced an increased olfactory sensitivity for the four odors used during the training, but also an overall increase of the olfactory function, whereas no anti-Parkinsonian treatment allowed this type of result. 

Finally, a recent study highlighted the effects of olfactory training on symptoms of depression [20]. In this study, the subjects followed an olfactory training over a period of five months or had to perform Sudoku daily for the control group. The odors used were the same as in Hummel’s study presented previously [21]. The results showed a significant decrease in the depression score (obtained with the Beck Depression Inventory) for the group who followed the five-month olfactory training compared to the group who performed Sudoku daily.

The mechanisms of action of odors’ effects on depression are not known yet. However, several hypotheses were developed: the volatile odorant compound could act as a pharmacological agent and enter the bloodstream. In such a situation, its effects will be dependent on the concentration of the compound [133]. Studies are still needed to understand the mechanisms of action of odors on behavior in animals or humans.

## 8. Conclusions

We have shown here the links between the olfactory system, depression, and quality of life. Different brain areas are involved in both depression and olfaction, and patients with depression regularly suffer from an impaired sense of smell. Further studies are needed to confirm that only olfactory threshold and hedonic perception are altered in depression while the odor identification capabilities are preserved. In the future, it is important to study the olfactory perception of depressed patients in a more natural environment reflecting everyday life and using more complex sensory (olfactory and gustatory) stimuli. These investigations could explain the role of olfactory impairment in the eating disorders frequently observed in depression.

In this review, we have also demonstrated that the presence of olfactory disorders can lead to a decrease in the quality of life of patients in several areas: food, social life, and work. Previous studies have demonstrated that other perceptual deficits affecting gustation [134], vision [135], or audition [136] may lead to depression by decreasing quality of life. There are no comparative studies investigating the importance of different perceptual sensory deficits on depressed patients’ quality of life. However, it is known that all senses participate in some sensory experiences like food intake. Such sensory experience will involve vision, audition, and kinesthesis, as well as the chemosensory modalities of olfaction, gustation, and chemesthesis that underlies flavor perception. However, sensory deficits are only partially assessed in some of the questionnaires evaluating the quality of life. Future improvement of the existing questionnaires are needed to create the tools suitable to the real problems of the clinical populations with olfactory deficits. Moreover, current clinical practice does not take into account olfactory impairments in depressed patients; olfactory deficits are not described in depression clinical criteria defined in DSM-5 [44]. The present review confirms the presence of sensory impairments and, specifically, the olfactory ones in the clinical spectrum of depression patients, and suggests that simple and inexpensive tools could be used to improve olfactory deficits. Furthermore, the current literature provides emerging evidence that olfactory stimulation (olfactory training) may be a promising tool for future therapeutic prospects. As suggested by Hummel et al. [21], it would be interesting to study the effects of this stimulation over time in order to know if the changes observed during olfactory training persist in time or not. Finally, studying the precise mechanisms of action of olfactory training is a must, in order to develop better olfactory training methods and to deepen knowledge about the olfactory system, depression, and how they affect quality of life.

## Figures and Tables

**Table 1 brainsci-08-00080-t001:** Brain areas involved in the processing of olfaction and depression.

Brain Areas	Olfaction		Depression	
	Observations	References	Observations	References
Olfactory bulb	Transmission of olfactory information to other brain areas	Brand, 2001 [50]	A bulbectomy causes changes (molecular and behavioral) similar to those of depression	Song and Leonard, 2005 [15]
			Reduced volume observed in depression	Negoias et al. 2010 [63]
Amygdala	Activation during the evaluation of intensity, hedonic aspect and emotional memory related to odors	Pouliot and Jones-Gotman, 2008 [67]	Hyperactivated in depression	Drevets, 2003 [68]
Hippocampus	Role in odor storage tasks	Kesner et al. 2002 [65]	Deficit in autobiographical memory	Lemogne et al. 2006 [66]
			Reduced volume observed in depression	Campbell et al. 2004 [33]
Orbitofrontal cortex	Controversial data: left part activated by unpleasant stimulus and right part activated by pleasant stimulus vs. activation not dependent of hedonic valence	Grabenhorst et al. 2011 [72] Zald and Pardo, 1997 [71]	Ventromedian part (hyperactivated) involved in rumination anxiety and sensitivity to pain and dorsal part (hypoactivated) psychomotor retardation, apathy, attention disorders, and working memory	Rogers et al. 2004 [69]
	Identification of odors and memorization	Zald et al. 2002 [73]		
Cingulate cortex	Activation dependent of hedonic valence of the odor	Fulbright et al. 1998 [76]	Reduced volume of anterior part: partly responsible for the increased recurrence of depressive episodes	Phillips et al. 2003 [74]
				van Tol et al. 2010 [35]
				Bhagwagar et al. 2008 [75]
Insula	Odor quality coding	Veldhuizen et al. 2010 [78]	Increased of activity during states of rest can be involved in inabilitiy to disengage from external events and lead to rumination	Sliz and Hayley, 2012 [79]
Habenula	Transfer olfactory information to other brain areas	Da Costa et al. 1997 [81]	Metabolic activity increased	Shumake et al. 2003 [80]
	Activated in response to emotionally-negative stimuli	Hikosaka et al. 2008 [82]		

**Table 2 brainsci-08-00080-t002:** Summary of publications on olfactory function in depression.

Study	Number of Patients Female (f)/Male (m)	Mean Age ± Standard Deviation	Measures and Tools	Conclusions
Swiecicki et al. (2009) [89]	D: 46 (f) C: 30 (f)	D: 38.2 ± 1.6 C: 35.4 ± 2.1	One testing session	
			Sniffin’ Stick Test:	
			Odor threshold	Intact odor threshold
			Odor identification	Intact identification abilities
			Odor characterization: pleasant/unpleasant/neutral	Intact odor hedonicity
Scinska et al. (2008) [88]	D: 25 (17/7) C: 60 (38/22)	D: 67.2 ± 1.2 C: 66.7 ± 0.9	One testing session	
			Sniffin’ Stick Test:	
			Odor threshold (n-butanol)	Intact odor threshold
			Odor identification	Intact identification abilities
Postolache et al. (1999) [87]	D: 24 C: 24	D: 42.8 ± 9.7 C: 42.1 ± 11.8	One testing session	
			Odor threshold: single stair-case procedure (PEA)	Intact odor threshold
			Identification: L-UPSIT and R-UPSIT	Intact identification abilities
Negoias et al. (2010) [63]	D: 21 (17/4) C: 21 (15/6)	D: 36.9 ± 10.1 C: 39.6 ± 11.4	One testing session	
			Sniffin’ Stick test:	
			Odor threshold (PEA)	Modified odor threshold
			Odor discrimination	Intact odor discrimination ability
			Odor identification	Intact odor identification ability
Lombion-Pouthier et al. (2006) [85]	D: 49 (35/14) C: 58 (36/22)	D: 43.4 ± 17.5 C: 38.4 ± 13.9	One testing session	
			Olfactory test (Ezus)	
			Odor sensibility	Modified odor threshold
			Detection/evaluation abilities (16 odors)	Intact detection/evaluation abilities
			Odor Identification	Intact identification abilities
			Intensity rating	Intact odor intensity
			Hedonic rating	Modified odor hedonicity
Pause et al. (2001) [86]	D: 24 (15/9) C: 25 (15/9)	D: 48.4 ± 13.2 C: 44.2 ± 12.6	Two testing sessions (S1 and S2)	
			Odor threshold (PEA and eugenol)	Modified odor threshold (S1)
			Subjective odors rating (intensity and hedonicity, 10 odors)	Modified odor hedonicity (S1) and intact odor intensity (S1/S2)
Pentzek et al. (2007) [99]	C: 20 (15/5) C: 30 (24/6)	D: 73.4 ± 5.6 C: 77.1 ± 6.8	One testing session	
			Sniffin’ Stick Test	
			Odor identification	Intact identification abilities
McCaffrey et al. (2000) [98]	D: 20 (11/9) A: 20 (13/7)	D: 67.55 ± 7.29 A: 74.15 ± 7.86	One testing session	
			Pocket Smell Test	Intact identification abilities
Solomon et al. (1998) [100]	D: 20 (13/7) A: 20 (12/8)	D: 69.4 ± 7.7 A: 74.5 ± 7.7	One testing session	
			Pocket Smell test	Intact identification abilities
Kopala et al. (1994) [97]	D: 21 (13/8) C: 77 (47/30)	D: 37.0 ± 9.6 C: 32.5 ± 11.1	One testing session	
			UPSIT	
			Odor identification	Intact identification abilities
Amsterdam et al. (1987) [96]	D: 51 (34/17) C: 51 (34/17)	D: 43 ± 13 (f); 49 ± 14 (m) C: age-matched	One testing session	
			UPSIT	
			Odor identification	Intact identification abilities
Naudin et al. (2012) [46]	D: 20 (12/6) C: 54 (36/18)	D: 50.1 ± 13.3 C: 49.5 ± 12.5	Two testing sessions (during acute phase of depression and after 6 weeks of antidepressant treatment)	
			Eight odors (four pleasant, two unpleasant, two neutral)	
			Hedonic rating	Modified odor hedonicity (S1) and improvement in (S2)
			Familiarity rating	Modified odor familiarity (S1, S2 for vanillin)
			Identification of single odors	Intact identification abilities of single odors (S1, S2)
			Odor intensity discrimination	Modified odor intensity discrimination (S1, S2)
			Odor identification abilities in binary mixture (one pleasant/one unpleasant)	Modified identification abilities in binary mixture) (S1, S2)
Zucco and Bollini (2011) [94]	Mild D: 12 (6/6) Severe D: 12 (6/6) C: 12 (6/6)	Mild D: 41.3 ± 6.4 Severe D: 41.9 ± 6.2 C: 39.8 ± 7.1	One testing session	
			Odor identification	Modified odor identification abilities
			Odor recognition memory (10 targets odors/30 distractors odors)	Modified odor recognition memory
Atanasova et al. (2010) [95]	D: 30 (12/18) C: 30 (12/18)	D: 34.6 ± 11.1 C: 33.4 ± 9.9	One testing session	
			Two odors: vanillin/butyric acid	
			Intensity rating	Modified odor intensity
			Odor intensity discrimination	Modified odor intensity discrimination
			Odor identification in binary mixture	Modified identification abilities in binary mixture
			Hedonics rating (single/binary mixture)	Modified odor hedonicity
			Odor discrimination	Intact odor discrimination ability for unpleasant odor and modified for pleasant odor
Clepce et al. (2010) [92]	D: 37 (21/16) C: 37 (21/16)	D: 48.3 ± 11.9 (m); 47.5 ± 11.3 (f) C: age-matched	Two testing sessions (during acute phase of depression and six months after the 1st phase)	
			Sniffin’ Stick Test:	
			Odor identification	Modified odor identification abilities (S1)
			Hedonics and intensity rating	Intact odor hedonicity and intensity (S1, S2)
Pollatos et al. (2007) [91]	D: 48 (34/14)	28.2 ± 5.8	One testing session	
			Sniffin’ Stick test:	
			Odor threshold	Modified odor threshold
			Odor discrimination	Intact odor quality discrimination
Croy et al. (2014) [27]	D: 27 (f) C: 28 (f)	D: 38.5 ± 10.6 C: 353 ± 10.3	Two testing sessions (before and after therapy)	
			Sniffin’ Stick test:	
			Odor threshold	Intact odor threshold (S1, S2)
			Odor discrimination	Modified odor discrimination (S1, S2)
			Odor identification	Intact odor identification (S1, S2)
Atanasova et al. (2012) [101]	D: 30 (12/18) C: 30 (12/18)	D: 36.6 ± 11.1 C: 33.4 ± 9.9	One testing session	
			Odor: vanillin/butyric acid	
			Hedonicity	Modified odor hedonicity
Naudin et al. (2014) [47]	D: 22 (16/6) C: 41 (24/17)	D: 33,2 ± 11.2 C: 34 ± 11	Two testing sessions (one during acute phase of depression and 100 days ± 64 after the 1st phase)	
			Eight odors	
			Hedonicity, Familiarity and emotion’ intensity rating	Modified odor hedonicity (S1; restore in S2), intact odor familiarity (S1, S2) and modified intensity of emotion (S1)
Naudin et al. (2014) [102]	D: 20 (15/5) A: 20 (14/6) C: 24 (17/9)	D: 64.9 ± 11.2 A: 73.0 ± 11.2 C: 67.0 ± 12.7	One testing session (in elderly)	
			Eight odors: four familiar and four unfamiliar	
			Odor recognition memory	Modified odor recognition memory for D and A
			Pleasantness, intensity, familiarity	Intact odors pleasantness, intensity and familiarity perception for D and A
Gross-Isseroff et al. (1994) [90]	D: 9 (8/1) C: 9 (8/1)	D: 49 ± 4.6 C: 49.1 ± 4.8	Three testing sessions (1st: acute phase of depression; 2nd: three weeks after the 1st and 3rd: sox weeks after the 1st)	
			Odors threshold (androsteron and isoamylacetate)	Intact odor threshold (1st and 2nd sessions) and modified odor threshold (3rd session)
Serby et al. (1990) [93]	D: 9 (m) C: 9 (m)	D: 50–59 years old C: age-matched	One testing phase	
			UPSIT (identification abilities)	Modified odor identification abilities
			Odor threshold (geraniol)	Intact odor threshold
Thomas et al. (2002) [103]	D: 16 C: 16	*i.i.*	Odor threshold	Intact odor threshold
			Intensity and familiarity ratings	Intact intensity and familiarity ratings abilities
Postolache et al. (2002) [104]	SAD: 14 (7/7) C: 16 (9/7)	SAD: 42.3 ± 11.5 C: 39.0 ± 10.0	One testing session	
			Odor threshold	Intact odor threshold
Oren et al. (1995) [105]	SAD: 21 (16/9) C: 21 (16/9)	SAD: 38 ± 9 C: 38 ± 9	One testing session	
			Odor identification	Intact odor identification abilities
Kamath et al. (2017) [106]	BPI: 43 (28/15) BPII: 48 (29/19) D: 134 (97/34) Anxiety: 48 (36/12) C: 72 (30/42)	BPI: 43.66 ± 15.6 BPII: 44.85 ± 15.6 D: 50.31 ± 14.87 Anxiety: 48.78 ± 17.12 C: 50.50 ± 18.09	One testing session	
			UPSIT (identification abilities)	Modified odor identification abilities (BPI, D)
			Sniffin’ Stick Test: Discrimination abilities and odor threshold	Intact discrimination abilities and odor threshold

A: Alzheimer patients; C: control group; D: depressed patients; SAD: seasonal affective disorder; *i.i.*: incomplete information; BP: bipolar depression; PEA: phenyl ethyl alcohol; UPSIT: University of Pennsylvania Smell Identification Test, S1: testing session during acute phase of depression, S2: testing session during remission and/or after therapy.

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
