# Peer review of "Depression, Olfaction, and Quality of Life: A Mutual Relationship"

_brainsci, 2018, doi:10.3390/brainsci8050080_

Round 1

Reviewer 1 Report

Overall this is an interesting topic area that warrants a critical review of the literature. This review would benefit from better organization and cohesion of topic areas. My suggestions for improvement are below.

For Section 1, it may be more helpful to have the paper start with an overview of the link between depression and olfaction and then have sections on behavioral olfactory findings, electrophysiological findings, neuroimaging studies/animal work in depression. The last section could focus on a mechanism for olfaction and depression.

Stating that depression is “associated with mood, emotional, and cognitive symptoms” seems circular (Line 53-54). It would be more helpful to state what the core symptoms of major depression are and to then describe the typical cognitive problems seen in depressed individuals.

For section 2.1, the authors state that olfaction is the most emotional sense but is there evidence comparing olfactory stimuli to gustatory, visual or auditory stimuli to strengthen this assertion. Also, a citation is missing for line 82.

For section 2.2, the relevance of this section to the overall aim of the paper is not clear, particularly the part regarding head trauma. It would be best to shorten this section and possibly incorporate it towards the end of section 2.3. 

For Table 2, there are studies missing from the publication list. The authors are encouraged to review the following paper to include a more complete listing of publications.

Psychopathology. 2013;46(2):63-74. doi: 10.1159/000338717. Epub 2012 Aug 7.

Olfaction in affective and anxiety disorders: a review of the literature.

Burón E., Bulbena A.

There are also more recent reviews by Croy (2017), Kohli (2016) and Taalman (2017) for references and several recent papers on olfaction and olfactory bulb volume findings in depression from early 2018 that would make this work more comprehensive. It would be helpful for the authors to state how the aim of this paper is different from the above mentioned reviews in the aims section of the paper.

Consider incorporating the below article in Section 3.1:

Li Q, Yang D et al (2015) Reduced amount of olfactory receptor neurons in the rat model of depression. Neurosci Lett 603:48–54

In Section 3.2 beginning on line 257, it would be helpful for the authors to discuss intensity ratings of odors to place the hedonic findings into context. Are there any studies of other types of ratings like edibility, familiarity, etc.?

How do olfactory findings in unipolar depression compare to what is seen in bipolar disorder or schizoaffective disorder?

Section 4, can the quality of life aspect be integrated sooner? The introduction of this section seems disjointed from the rest of the paper.

Author Response

Response to Reviewer 1,

Thank you very much for the given opportunity to improve the quality of our manuscript .

We made all minor changes and corrections in the new version of the manuscript and provide a point-by-point response below. The changes made in this revised document are highlighted in yellow.

Yours sincerely,

On behalf of the authors of the manuscript,

Responses

Suggestion:

For Section 1, it may be more helpful to have the paper start with an overview of the link between depression and olfaction and then have sections on behavioral olfactory findings, electrophysiological findings, neuroimaging studies/animal work in depression. The last section could focus on a mechanism for olfaction and depression.

Response:

As suggested by the reviewer, we described in section 1, the link between depression and olfaction. We also added a brief description of the link of these two chronic affections and the quality of live in order to response to another suggestion of the reviewer; “Section 4, can the quality of life aspect be integrated sooner? The introduction of this section seems disjointed from the rest of the paper.” (p. 1-2, lines 25-68).

The order of the other sections has not be changed to keep a logical connection of the approached topics: the etiologies of depression, functioning of the olfaction and evaluation of olfactory function the anatomic link between olfaction and depression….

Suggestion:

Stating that depression is “associated with mood, emotional, and cognitive symptoms” seems circular (Line 53-54). It would be more helpful to state what the core symptoms of major depression are and to then describe the typical cognitive problems seen in depressed individuals.

Response:

This has been corrected in the new version of the manuscript (p. 2-3, lines 94-103).

Suggestion:

For section 2.1, the authors state that olfaction is the most emotional sense but is there evidence comparing olfactory stimuli to gustatory, visual or auditory stimuli to strengthen this assertion. Also, a citation is missing for line 82.

Response:

This point has been clarified in the new version of the manuscript and a citation was added (p. 3, lines 122-125).

Suggestion:

For section 2.2, the relevance of this section to the overall aim of the paper is not clear, particularly the part regarding head trauma. It would be best to shorten this section and possibly incorporate it towards the end of section 2.3.

Response:

As suggested by the reviewer, we shorted the section concerned the origin of olfactory disorders and incorporated it towards the end of section “Evaluation of olfactory functions” (p. 4, lines 173-183).

Suggestion:

For Table 2, there are studies missing from the publication list. The authors are encouraged to review the following paper to include a more complete listing of publications.

Psychopathology. 2013;46(2):63-74. doi: 10.1159/000338717. Epub 2012 Aug 7.

Olfaction in affective and anxiety disorders: a review of the literature.

Burón E., Bulbena A.

Response:

As suggested, we added more studies in Table 2, after reviewing the paper of Burón and Bulbena (2011) (see Table 2).

Suggestion:

There are also more recent reviews by Croy (2017), Kohli (2016) and Taalman (2017) for references and several recent papers on olfaction and olfactory bulb volume findings in depression from early 2018 that would make this work more comprehensive. It would be helpful for the authors to state how the aim of this paper is different from the above mentioned reviews in the aims section of the paper.

Response:

As suggested by the reviewer, we read all recent reviews (Croy (2017), Kohli (2016) and Taalman (2017)) and incorporated them in the new version of the manuscript (reference n°13, 18 and 26). We also stated how the aim of our paper is different from the above mentioned reviews in the aims section of the paper (p. 2, lines 61-68).

Suggestion:

Consider incorporating the below article in Section 3.1:

Li Q, Yang D et al (2015) Reduced amount of olfactory receptor neurons in the rat model of depression. Neurosci Lett 603:48–54

Response:

The suggested article (Li Q, Yang D et al, 2015) has be incorporated in the manuscript; Section 4.1 (p. 4, lines 195-198).

Suggestion:

In Section 3.2 beginning on line 257, it would be helpful for the authors to discuss intensity ratings of odors to place the hedonic findings into context. Are there any studies of other types of ratings like edibility, familiarity, etc.?

Response:

These points have be discussed in the new version of the manuscript (p. 7, lines 284-294).

Suggestion:

How do olfactory findings in unipolar depression compare to what is seen in bipolar disorder or schizoaffective disorder?

Response:

This point has be discussed in the new version of the manuscript (p. 10-11, lines 300-310).

Suggestion:

Section 4, can the quality of life aspect be integrated sooner? The introduction of this section seems disjointed from the rest of the paper.

Response:

As suggested by the reviewer, the quality of life aspect has be integrated sooner; in the introduction section, (p. 1-2, lines 26-37 and 61-63).

Reviewer 2 Report

This is an interesting review in an area that needs investigating. This paper also has good consideration. However, cited references are entirely old. For example, there is a review of Kohli P et al.( Chem Senses 2016) on the correlation between the olfactory disturbance and the severity of depression. If it does not become a review without using old papers, it also means that research is not progressing. Although it is an interesting theme, I hope that the meaning to review now becomes clearer.

It is good to quote a paper on olfactory bulbectpmy, but for rodents without cerebral development olfactory bulbectpmy is a procedure that can be the destruction of the brain, there is a great limit to applying findings to human. A discussion on whether olfactory bulbectomy is valid as a depression model is necessary.

If the influence of olfaction on QOL is discussed in relation to depression, it is necessary to clarify how the difference between other senses and QOL in relation to depression is different.

Author Response

Response to Reviewer 2,

Thank you very much for the given opportunity to improve the quality of our manuscript.

We made all minor changes and corrections in the new version of the manuscript and provide a point-by-point response below. The changes made in this revised document are highlighted in yellow.

 Yours sincerely,

On behalf of the authors of the manuscript,

Responses

Suggestion:

This is an interesting review in an area that needs investigating. This paper also has good consideration. However, cited references are entirely old. For example, there is a review of Kohli P et al.( Chem Senses 2016) on the correlation between the olfactory disturbance and the severity of depression. If it does not become a review without using old papers, it also means that research is not progressing. Although it is an interesting theme, I hope that the meaning to review now becomes clearer.

Response:

As suggested by the reviewer, we read several recent reviews (Croy (2017), Kohli (2016) and Taalman (2017)…) and incorporated them in the new version of the manuscript (see section “Introduction”). We also used theses reviews to complete the table 2.

Suggestion:

It is good to quote a paper on olfactory bulbectpmy, but for rodents without cerebral development olfactory bulbectpmy is a procedure that can be the destruction of the brain, there is a great limit to applying findings to human. A discussion on whether olfactory bulbectomy is valid as a depression model is necessary.

Response:

This point has be discussed in the new version of the manuscript (p. 12, lines 397-400).

Suggestion:

If the influence of olfaction on QOL is discussed in relation to depression, it is necessary to clarify how the difference between other senses and QOL in relation to depression is different.

Response:

This point has be discussed in the conclusion of new version of the manuscript (p. 13, lines 441-447).
